# A Modified Single-Frequency PPP Method for the Positioning and Time Transfer with BDS-3

**DOI:** 10.3390/s22239059

**Published:** 2022-11-22

**Authors:** Mingjun Ouyang, Xiangwei Zhu, Junzhi Li, Yang Liu

**Affiliations:** 1School of Electronics and Communication Engineering, Sun Yat-Sen University, Guangzhou 510006, China; 2School of Aeronautics and Astronautics, Sun Yat-Sen University, Guangzhou 510006, China

**Keywords:** BDS-3, time transfer, precise point positioning, single frequency, group and phase ionospheric correction (GRAPHIC)

## Abstract

In this paper, time–frequency transfer and positioning experiments with signal coexistence in the BDS system were conducted using the four types of open service signals of the BDS-3 satellite (B1I, B1C, B2a, and B3I), as well as the B2I signals broadcast by the BDS-2 satellites. The experiments used the single-frequency PPP (precise point positioning) method. The experiment validated a modified version of the group and phase ionospheric correction (GRAPHIC) technique. The results demonstrate that, with a single frequency, 18 selected stations may provide positioning results accurate to within a few decimeters. The positioning accuracy of five frequencies signals is improved by 40.4%, 32.2%, 80.3%, 12.4%, and 10.3% when compared to the positioning accuracy of the same signals when using the general observation approach. Currently, the frequency stability may be as precise as dual frequencies with BDS.

## 1. Introduction

The BDS-3’s construction process began in 2015. The BDS-3 includes 3 GEO (Geostationary Earth Orbit), 24 MEO (Medium Orbit Earth Satellite), and 3 IGSO (Inclined Geosynchronous Orbit) satellites, which can provide worldwide coverage in contrast to the regional service offered by the BDS-2 [1]. Numerous studies on the BDS-3’s determination of satellite orbit, estimation of clock difference, location, time and frequency transfer, and other relevant topics have been carried out both during and after the launch of various BDS-3 experimental satellites.

By utilizing the MGEX (The Multi-GNSS Experiment and Pilot Project) and iGMAS (International GNSS Monitoring & Assessment System) stations, Li et al. realized the BDS-3’s determination of satellite orbit in 2018 [2]. A statistical analysis of the new signals (B1I, B1C, B2a, B2b, and B3I) produced by the BDS-3 in terms of multipath, signal-to-noise ratio, and other factors was conducted by Zhang et al. [3]. Using B1C and B2a frequency signals, Ye et al. calculated the BDS-3’s satellite orbit [4]. The precise clock difference and orbit of the BDS-3 were established by Yan et al. [5]. By examining a BDS-3 satellite in the L and Ka bands, Xie et al. determined the orbit [6]. Yang et al. provided a detailed introduction to the coordinate system and overall design of the BDS-3 system. They then presented the results of the signal-in-space ranging errors (SISRE) through measurements taken over the course of four days by eight BDS-3 satellites and contrasted the broadcast ephemeris and precise orbit measurements. According to the findings, the three-dimensional (3D) satellite orbit’s root mean square (RMS) was approximately 1.07 m, the satellite clock difference’s average RMS was 1.12 ns, and the average SISRE was 0.44 m [7]. The precision level of the single point positioning (SPP) of the new BDS-3 signal, according to research by Zhang et al., was determined to be comparable to that of the GPS [8].

Ge et al. evaluated the performance of the BDS-2’s dual-frequency ionosphere-free (IF) combination in terms of time–frequency transfer by using the iGMAS product [9]. Tu et al. found that there was less observation noise and they could achieve the same accuracy of time transfer as the GPS by using the triple-frequency undifferenced and uncombined (UC) [10]. Zhang et al. evaluated the GPS PPP using different institutional products (IGS, GFZ, CODE), and obtained a sub-nanosecond time transfer accuracy [11]. Ge et al. evaluated the time–frequency transfer of the BDS-3 through experiments with different combinations [12], and reached the conclusion that the stability and accuracy of the new signals perform better.

However, all of the mentioned research, whether it be focused on orbit determination and clock difference computation, positioning, or application research of time–frequency transfer, was based on the multi-frequency signals of BDS-3 and neglected to take into account the needs of single-frequency users, who currently make up the majority of GNSS users. To address consumer navigation and positioning needs, GNSS navigation processors are currently available in almost all smartphones. The depth and extent of the consumer market’s applications can be increased and enhanced by assessing the performance of a single-frequency method.

BDS-2 and BDS-3 coexist in the sky presently. When used for positioning or time transfer, BDS-2 and BDS-3 can be regarded either as the same system or two different systems. The BDS-3 was employed in the program design as a different system from the BDS-2 in this study, and BDS-2 served as a benchmark. In this study, a modified single-frequency GRAPHIC model is proposed. Between the code and carrier phase observations, an ionosphere-free technology is developed because the ionospheric refraction corrections of the code’s pseudo-range and carrier phase observations are identical in size but opposite in sign. This study of location and time–frequency transfer is based on the two single-frequency model algorithms (single-frequency UC model and single-frequency modified GRAPHIC model), and it assesses the effectiveness of the existing single-frequency coexistence of BDS-2 and BDS-3 using the PPP technique.

The remainder of this study is structured as follows: The current BDS-3 state is explained in full in Section 2. The two primary single-frequency algorithms employed in this work are described in Section 3. The specific processing plan and data selection are presented in Section 4. The results are analyzed and compared in Section 5. The relevant conclusions are summarized in Section 6.

## 2. Current BDS-3 Status

The BDS transitions from the second generation of the regional system to the third generation of the global system in accordance with a three-step approach. On 23 June 2020, the final BDS-3 networking satellite was successfully launched, completing the third generation’s construction, and transforming the BDS into a global satellite navigation system.

Currently, the BDS-2 system’s 15 satellites run steadily and continually. Before the BDS-3 system was fully networked, five experimental BDS-3 satellites were launched and validated by on-orbit tests. The performance and service life of the satellite were significantly improved by the development of higher-performance rubidium atomic clocks (with daily stability of 1 × 10^−13^ magnitude) and hydrogen atomic clocks (with daily stability of 1 × 10^−14^ magnitude). The deployment of the basic system constellation was successful, and a stable and dependable intersatellite link was constructed. The integration of navigation and positioning, communication, and improved services has been made possible by BDS-3. It offers six different types of services in addition to navigation and positioning, including satellite-based enhancement, broadcasting accurate positioning information, communication, and international search and rescue [1].

In order to provide public navigation signals with four frequencies, BDS-3 is downward compatible with BDS-2 B1I and B3I signals. Additionally, two new signals, B1C and B2a (compatible with GPS L1/L5 and Galileo E1/E5A), are introduced. Compared to conventional signals, B1C and B2a signals have a broader bandwidth, greater range accuracy, and better compatibility. LDPC (low-density parity-check) channel coding is typically employed to enhance the demodulation performance of weak signals, and pilot channels are typically included to boost the receiving sensitivity of weak signals (as shown in Table 1). Additionally, B1C and B2a adopt new orbit parameters, BGCNAV1 and BGCNAV2 navigation messages, and the BDGIM (Beidou global ionospheric delay correction model) based on a spherical harmonic model. Compared to BDS-2, the accuracy of orbit and ionosphere correction has been dramatically improved [7]. BDS-2 and BDS-3 currently cohabit because of the long service life of the BDS-2 satellite. Due to the importance of this topic to the majority of consumers, this article examines the use of the single-frequency PPP method in the coexistence state of two generations of systems.

## 3. Principle and Improvement of Single-Frequency PPP Algorithm

### 3.1. Basic Formula

The observation data can be expressed as follows:(1)$$P_{r,j}^{s}=\rho_{r}^{s}+c\cdot dt_{r}^{s}-c\cdot dt^{s}+MF_{w}(e)\cdot Z_{w}+\gamma_{j}^{s}\cdot I_{r,1}^{s}+(d_{r,j}^{s}-d_{j}^{s})+\varepsilon_{r,j}^{s}(P)$$
(2)$$\Phi_{r,j}^{s}=\rho_{r}^{s}+c\cdot dt_{r}^{s}-c\cdot dt^{s}+MF_{w}(e)\cdot Z_{w}-\gamma_{j}^{s}\cdot I_{r,1}^{s}+\lambda_{j}^{s}\cdot (N_{r,j}^{s}+b_{r,j}^{s}-b_{j}^{s})+\varepsilon_{r,j}^{s}(\varphi )$$
where *r*, *j*, and *s* stand for receivers, frequency, and satellite systems, respectively; $p_{r,j}^{s}$ and $\Phi_{r,j}^{s}$ represent the pseudo-range observation and the carrier phase observation, respectively; $\lambda_{j}^{s}$ denotes the wavelength of frequency *j*; $\rho_{r}^{s}$ represents the geometric distance between receiver and satellite; *c* represents the speed of light; $dt_{r}^{s}$ and $dt^{s}$ denote the receiver clock difference and the satellite clock difference, respectively; *e* represents the elevation angle of a satellite; $MF_{w}(e)$ denotes the wet delay projection function; $Z_{w}$ refers to the wet delay of the zenith direction; $I_{r,1}^{s}$ represents the tilt direction delay of the ionosphere; multiple factor $\gamma_{j}^{s}={\left(f_{1}^{s}/f_{j}^{s}\right)}^{2}$ is based on frequency; $d_{r,j}^{s}$ and $d_{j}^{s}$ represent uncalibrated code delays (UCDs) at the receiver and satellite, respectively; $N_{r,j}^{s}$ represents the integer ambiguity, $b_{r,j}^{s}$ and $d_{j}^{s}$ denote uncalibrated carrier phase delays (UPDs) at the receiver and satellite, respectively; lastly, $\varepsilon_{r,j}^{s}(P)$ and $\varepsilon_{r,j}^{s}(\varphi )$ represent the carrier phase noise and the pseudo-range noise, respectively.

### 3.2. Single-Frequency UC PPP Model 

The first-order ionosphere delay has been removed for the dual-frequency ionosphere-free combination, which means that more than 99.9% of the ionospheric error has been eliminated. The ionosphere’s delay can only be utilized as a parameter for an estimate, a model that has to be adjusted, or external precision products for a particular frequency.

The satellite clock difference of precision products is estimated using the combination of P1 and P2 pseudo-ranges. Therefore, the satellite clock ($d_{IF12}^{s}$) represents a combination of satellite UCDs (($d_{IF12}^{s}$)) and actual clock difference $dt^{s}$, which is expressed as follows:(3)$$c\cdot dt_{IF12}^{s}=c\cdot dt^{s}+(\alpha_{12}^{2}\cdot d_{1}^{s}+\beta_{12}^{2}\cdot d_{2}^{s})$$
where $\alpha_{12}^{s}=\frac{\left(f_{1}^{s}\right)}{{\left(f_{1}^{s}\right)}^{2}-{\left(f_{2}^{s}\right)}^{2}}$ and $\beta_{12}^{s}=-\frac{\left(f_{2}^{s}\right)}{{\left(f_{1}^{s}\right)}^{2}-{\left(f_{2}^{s}\right)}^{2}}$.

The precise clock difference of the BDS corresponds to the B1I/B3I ionosphere-free combination, and the clock difference from broadcasted ephemeris of the BDS corresponds to the B3I. Therefore, for the single-frequency or other mixed-frequency user, TGD (time group delay) or DCB (differential code bias) should be utilized to be corrected, as shown in Figure 1. The particular corrective techniques are discussed in [15,16]. The precise corrected products of the Chinese Academy of Sciences’ Institute of Geodesy and Geophysics (IGG), which includes the GPS, GLONASS, Galileo, BDS, and QZSS, provided the DCB correction for this investigation with an accuracy range of 0.2 ns to 0.6 ns [17].

The COD precision clock product was utilized in this study. Equations (1) and (2) are linearized using the precision clock difference product, and Formula (3) is then substituted into Equation (4):(4)$$\left\{ \begin{array}{l} p_{r,1}^{s}=\mu_{r}^{s}\cdot x+c\cdot dt_{r}+d_{r,1}^{s}+MF_{w}(e)\cdot Z_{w}+I_{r,1}^{s}-\beta_{12}^{s}\cdot DCB_{P1P2}^{s}+\varepsilon_{r,1}^{s}(P)\\ l_{r,1}^{s}=\mu_{r}^{s}\cdot x+c\cdot dt_{r}-I_{r,1}^{s}+d_{IF12}^{s}+MF_{w}(e)\cdot Z_{w}+\lambda_{1}^{s}\cdot \left(N_{r,1}^{s}+b_{r,1}^{s}-b_{1}^{s}\right)+\varepsilon_{r,1}^{s}(\varphi ) \end{array}\right.$$
where $p_{r,1}^{s}$ and $l_{r,1}^{s}$ represent the calculated value of the observation value minus the pseudo-range and carrier phase observation, respectively; $\mu_{r}^{s}$ is the unit vector between the receiver and satellite; x denotes the position vector relative to a prior coordinate position; and the other parameters are the same as those defined for Equations (1) and (2).

### 3.3. Modified GRAPHIC PPP Model

Yunck put forth the GRAPHIC approach in 1993 [18]. The noise of the equation is decreased by half, and the rate of convergence of the PPP is sped up by 20% by using the physical features of ionospheric delay and the carrier phase, which are identical in size but opposite in sign, to eliminate the influence of ionospheric delay [19]. Equations (1) and (2) allow for the following expression of the single PPP model’s modified GRAPHIC:(5)$$\left\{ \begin{array}{l} L_{r,1}^{s}=1/2\cdot (p_{r,j}^{s}+l_{r,1}^{s})=\mu_{r}^{s}\cdot x_{r,1}^{s}+c\cdot dt_{r}+MF_{w}(e)\cdot Z_{w}+1/2\cdot \lambda_{1}^{s}\cdot N_{r,1}^{s}+B_{r,1}^{s}+\varepsilon_{r,1}^{s}(L)\\ p_{r,1}^{s}=\mu_{r}^{s}\cdot x+c\cdot dt_{r}+d_{r,1}^{s}+MF_{w}(e)\cdot Z_{w}+I_{r,1}^{s}-\beta_{12}^{s}\cdot DCB_{P1P2}^{s}+\varepsilon_{r,1}^{s}(P) \end{array}\right.$$
(6)$$B_{r,1}^{s}=1/2\cdot d_{r,1}^{s}+1/2\cdot d_{IF12}^{s}-1/2\cdot \beta_{12}^{s}\cdot DCB_{P_{1}P_{1}}^{s}+1/2(b_{r,1}^{s}-b_{1}^{s})$$

The pseudo-range observation equation is added to the equation in order to prevent rank deficit, and the ionospheric delay is fixed by employing ionospheric products in this equation. The parameters for the clock difference are sorted, and are denoted by:(7)$$\left\{ \begin{array}{l} c\cdot d{\bar{t}}_{r}=c\cdot dt_{r}+B1_{r,1}^{s}\\ {\bar{N}}_{r,1}^{s}=\lambda_{1}^{s}\cdot N_{r,1}^{s}+\delta B_{r,1}^{s} \end{array}\right.$$
where $B1_{r,1}^{s}$ represents the average delay, and $\delta B_{r,1}^{s}$ denotes the delay of the satellite, which is absorbed into the ambiguity parameter and the satellite clock difference parameter. The final estimated parameter vector $\bar{X}$ is expressed as:(8)$$\bar{X}=\left[\begin{array}{cccc} x & d{\bar{t}}_{r} & Z_{w} & {\bar{N}}_{r,1}^{s} \end{array}\right]$$

It should be noted that in the newly constructed function model, the pseudo-range and GRAPHIC equations in (5) and (6) both used pseudo-range observations, so there is a correlation between the combined observations. In this paper, a stochastic model is constructed based on the covariance propagation law. Suppose ${\hat{\sigma }}_{P_{1}}={\hat{\sigma }}_{P_{2}}={\hat{\sigma }}_{P},\;{\hat{\sigma }}_{\Phi_{1}}={\hat{\sigma }}_{\Phi_{2}}={\hat{\sigma }}_{\Phi }$, then the corresponding satellites are obtained from the error propagation law, and the values of each position of the covariance matrix are:(9)$$\left\{ \begin{array}{l} P_{IF,Li}=0.5(P_{i}+\Phi_{i})\\ Q_{Pi,Pi}={\hat{\sigma }}_{P}^{2}\\ Q_{UofC,UofC}=0.25{\hat{\sigma }}_{P}^{2}+0.25{\hat{\sigma }}_{\Phi }{}_{}^{2}\\ Q_{Pi,UofC}=0.5{\hat{\sigma }}_{P} \end{array}\right.$$

## 4. Data Selection and Processing

### 4.1. Data Selection

BDS-3 broadcasts can be tracked by a few MGEX tracking stations. This research chooses 18 MGEX sites for the positioning experiment, as indicated in Figure 2, to test the viability of a single-frequency algorithm and to imitate the performance of global distribution. For the high-precision time–frequency transfer study, some of the stations have connected high-precision atomic clocks.

The data were collected on April 9, 2020 (DOY 100, YEAR2020), and the sampling rate was 30 s. The public service data (B1I, B1C, B2I, B2a, and B3I) of the BDS-3 and BDS-2 could be received. Three stations, CUSV, HARB, and WTZZ, were selected for the time transfer experiment with the external atomic clock. Two time links, WTZZ-HARB and WTZZ-CUSV, were formed with WTZZ as the center.

### 4.2. Strategy Design and Filtering Algorithm

The single-frequency PPP UC strategy (referred to as Strategy 1) and the modified single-frequency GRAPHIC strategy (denoted as Strategy 2) are the two types of single-frequency PPP strategies that are devised and compared in this paper. Table 2 provides a comprehensive PPP algorithm technique.

## 5. Results and Analysis

The location of the current frequency served as the basis for a detailed comparison and analysis of the two single-frequency PPP techniques. Some of the graphs in the text below are marked with the UC method symbol, while others are marked with the modified GRAPHIC symbol.

### 5.1. Positioning Results

The BDS-3 is a global navigation system, in contrast to the BDS-2 system. The BDS-2’s visible satellite count and PDOP value of sites have been greatly modified in comparison, which means that it now meets the needs of all different kinds of users worldwide. Increased signal is useful for increasing the performance of the PDOP value outside the Asia Pacific region, according to Jiao et al. [23], who obtained the statistics of the PDOP value of the BDS-2 and BDS-3. On 23 June 2020, the final BDS-3 satellite was launched, forming a global satellite system. In a global sense, it is analogous to the GPS system in terms of satellite visibility and PDOP value. Due to the specific GEO and IGSO of the BDS, its satellite visibility and PDOP value will be superior to those of the GPS system even in the Asia Pacific, Australia, and some other regions. BDS-3 has 11 to 14 visible satellites in the 60° S to 60° N and 50° E to 170° E sectors, which is more than the GPS and Galileo systems, which have a range of 1 to 3 and 3 to 7, respectively.

The visible satellite number and PDOP value of the domestic station WUH2 employed in the experiment are displayed in Figure 3 at (DOY 001, 2020). As observed in Figure 3 and Figure 4, there were typically more than 16 visible satellites per day, and the PDOP value was approximately 2.2, which was able to better fulfill the needs of various applications, including location, time–frequency transfer, tropospheric solution, etc. To accomplish the continuousness and robustness of the technique, the user may find it advantageous to have a large number of redundant satellites and a good PDOP value.

In the experiment shown in Figure 5, the pairing of BDS BI1 and B3I dual-frequency in an ionosphere-free combination was used as a reference. COD was employed in this work as the accurate orbit and clock difference product. The satellite clock does not need to be altered by the DCB because its satellite clock difference product was based on the combination of B1I and B3I frequencies. The positioning results of the WHU2 station’s dual-frequency combination are shown in Figure 5, where it is clear that the accuracy in the horizontal directions E and N reached millimeter levels, while the accuracy in the vertical direction U was only marginally worse but eventually reached centimeter levels after convergence.

The results for station WUH2 as determined by the single-frequency UC (Strategy 1) and modified GRAPHIC (Strategy 2) strategies are shown in Figure 6. The findings demonstrate that, following convergence, the positioning accuracy was at the decimeter level in three directions (E, N, and U). Some results appeared to be influenced by systematic biases as a result of the absence of an ionosphere. Overall, U’s vertical direction accuracy is less accurate than its horizontal direction accuracy (N and E).

For five frequencies, the positioning stability and accuracy of Strategy 2 were superior to those of Strategy 1. The receiver decoded data improperly as a result of the B1C frequency being interrupted for two hours, which led to data loss.

A total of 18 stations (as depicted in Figure 2) dispersed across the globe were employed in the average statistical comparison in order to further verify and examine the aforementioned conclusions. The positioning outcomes for each frequency point of 18 stations using Strategy 1 and Strategy 2 are shown in Figure 7 and Figure 8, respectively. Figure 9 displays the total station data for 18 stations from each frequency’s perspective.

As presented in Figure 7 and Figure 8, the positioning accuracy of all stations and all frequencies, except for some stations with B1I/B2a/B2I signals, Strategy 2 was less than 0.5 decimeter, while that of the Strategy 1 was less than 0.75 decimeter. The poor performance of the B2I signal was due to the fewer number of visible satellites. Only BDS-2 transmitted B2I, and the number of satellites was less than four at some times in some areas. The positioning results of B1C and B3I were obviously better than other frequencies. The preliminary analysis showed that there were more satellites, and the data quality was better.

In Figure 9, the average position value of 18 stations is shown from the point of view of each frequency. The positioning results of Strategy 2 are better than those of Strategy 1, and the frequencies B1I, B1C, B2I, B2A, and B3I are increased by 40.4%, 32.2%, 80.3%, 12.4%, and 10.3%, respectively, compared with Strategy 1.

### 5.2. Comparison of Time and Frequency Transfer

The old and new single-frequency BDS signals (B1I, B1C, B2I, B2a, B3I) were tested with respect to high-precision time–frequency transfer. Three MGEX stations (CUSV, WTZZ, HARB) were selected, all of which were externally connected with high-precision atomic clocks. Based on the CUSV, two time transfer links (CUSV–WTZZ, CUSV–HARB) were formed. In [24], it has been proven that the ionosphere-free combination of B1I and B3I of BDS-3 can achieve an equivalent time transfer accuracy, compared with the GPS system. Therefore, the ionosphere-free combination of B1I and B3I of BDS was adopted as a reference. Due to the system’s deviation between single-frequency and dual-frequency combinations, the STD method was used for statistical comparison. In addition, the modified Allan variance was used for analysis and comparison of frequency stability of the time links.

$X_{i}^{}$ is the clock error or clock links obtained by the single-frequency method minus the reference clock error or clock links of the IF method. N is the number of all clock difference sequence points. $\bar{X}$ is average value of $X_{i}^{}$.
(10)$$\mathrm{STD}=\sqrt{\frac{\sum_{i=1}^{N}{\left(X_{i}-\bar{X}\right)}^{2}}{N}}$$

#### 5.2.1. Clock Difference Result

Figure 10 shows the clock difference statistics’ STD results of two strategies of three stations (CUSV, HARB, WTZZ). It can be seen that the clock difference statistics’ STD of Strategy 2 is significantly smaller than that of Strategy 1, which means that Strategy 2 is very helpful in improving clock error accuracy compared with Strategy 1.

Figure 11 shows that the STD of three stations can be stabilized at the sub-nanosecond order by using the single-frequency GRAPHIC method (Strategy 2). When Strategy 2 was used, the clock difference parameter was more stable and accurate than when the UC method (Strategy 1) was used. The performance of the new B1C signal was obviously better than other frequency signals. However, this is the result of only three stations, and more stations are needed to verify this result, and this is out of the scope of this paper. Due to the fewer number of B2I satellites, the performance of B2I is significantly worse than that of the other frequencies.

#### 5.2.2. Time Links Result

Figure 12 shows two time link sequences of two strategies and five frequencies. As shown in Figure 12, there was a systematic deviation between the time link of each frequency and the combination of the ionosphere-free correction. This systematic deviation originated from two aspects: (1) the difference between BDS-2 and BDS-3 systems; (2) DCB delay from the receiver and satellites: the satellite had been corrected, but the receiver had not been corrected. The sum of these two parts constitutes the system difference as shown in Figure 12.

Taking the ionosphere-free dual-frequency results of B1I and B3I as a reference, the single-frequency results of five frequencies were calculated by difference with each other to obtain the STD in Figure 13.

According to the link statistics, the following conclusion can be drawn:The new B1C signal is obviously better than other frequencies, while the advantage of the B2a signal is not obvious, which may be due to the fewer number of satellites;B3I performs well, which may be due to the large number of satellites. As we have stated, BDS-2 and BDS-3 can be used at the same time;B2I performance is worse than that of the other four frequencies, and the number of satellites is not enough. So, the performances of both links and strategies are worse.

The above is the analysis of time link accuracy. Next, the stability of corresponding time links is assessed.

Figure 14 show the statistical results of frequency stability of two strategy links: the CUSV–HARB and CUSV–WTZZ links. Based on these results, the following conclusions can be drawn:The frequency stability of two single-frequency links can reach the same accuracy as the one of dual-frequency combinations;As shown in Figure 14, The stabilities of B1I, B2I, and B2a frequencies of Strategy 1 are slightly better than those of Strategy 1 with an average time range from 100 s to 400 s. And we can see, five frequencies of Strategy 2 are closer to the reference value of B1I–B3I.

## 6. Conclusions

In this study, the time transfer and positioning tests for five types of single-frequency signals utilizing the PPP technique were conducted, using four types of open service signals (B1I, B1C, B2a, and B3I) of the BDS-3 and the B2I signals broadcast by the BDS-2. The standard UC model was criticized for its lack of flexibility, so a GRAPHIC combination with some tweaks to its stochastic underpinnings was offered as an alternative. In light of the fact that BDS-2 and BDS-3 are already in use together, this research intends to investigate the preferences of single-frequency users. The results of experiment show that:Eighteen globally distributed experimental stations can achieve sub-decimeter-level positioning accuracy with the BDS single-frequency static PPP method;Compared to the UC method, the modified GRAPHIC method improved the stability and accuracy significantly, and the accuracy of five frequencies (B1I, B1C, B2I, B2a, and B3I) was increased by 40.4%, 32.2%, 80.3%, 12.4%, and 10.3%, respectively, when positioning with single-frequency PPP method;The single-frequency clock difference STD can reach the level of 1 nanosecond; the time transfer link can achieve the STD of about 2 nanoseconds; and the frequency stability of the time transfer link can reach the same accuracy as the dual-frequency combination.

## Figures and Tables

**Figure 1 sensors-22-09059-f001:**
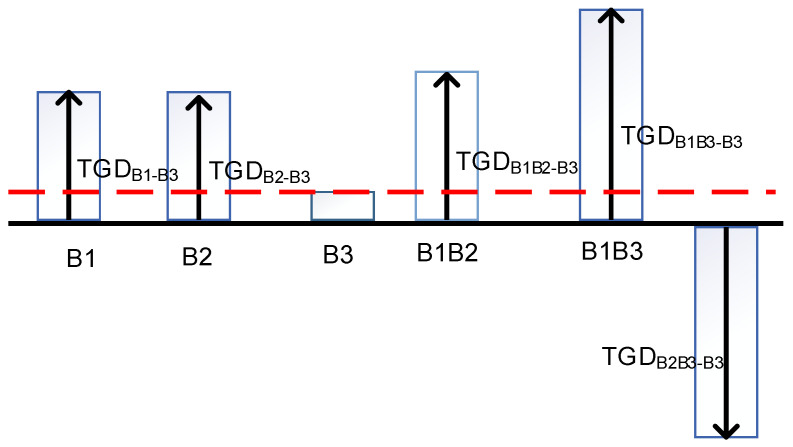
Presentation of the TGD relationship (This red line indicates that B3 is the benchmark).

**Figure 2 sensors-22-09059-f002:**
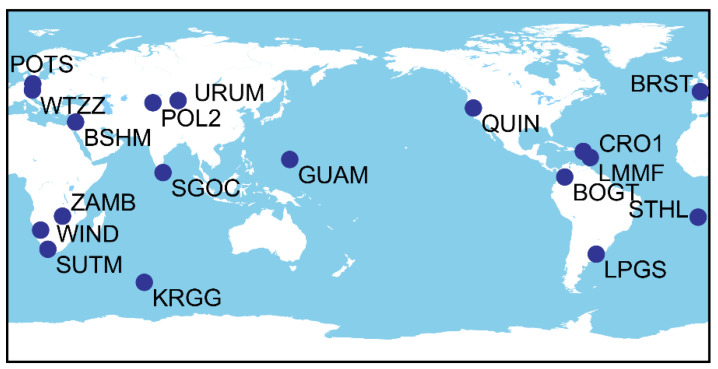
Distribution of selected stations.

**Figure 3 sensors-22-09059-f003:**
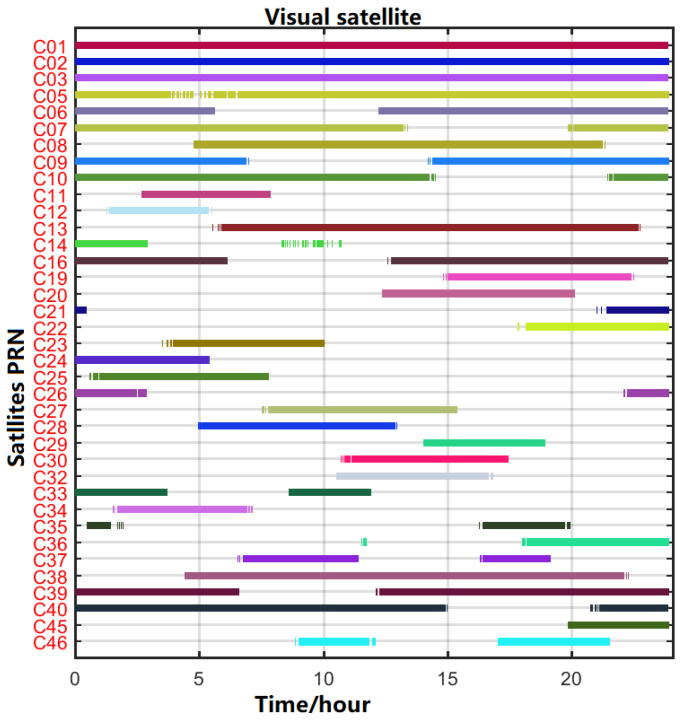
Satellite visibility of the WUH2 station and BDS satellite on 09 April 2020.

**Figure 4 sensors-22-09059-f004:**
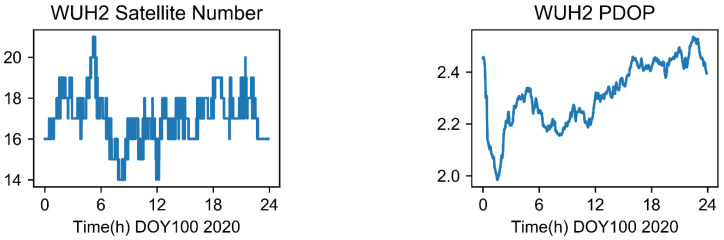
The visible satellite number and PDOP value of the WHU2 station and BDS satellite.

**Figure 5 sensors-22-09059-f005:**
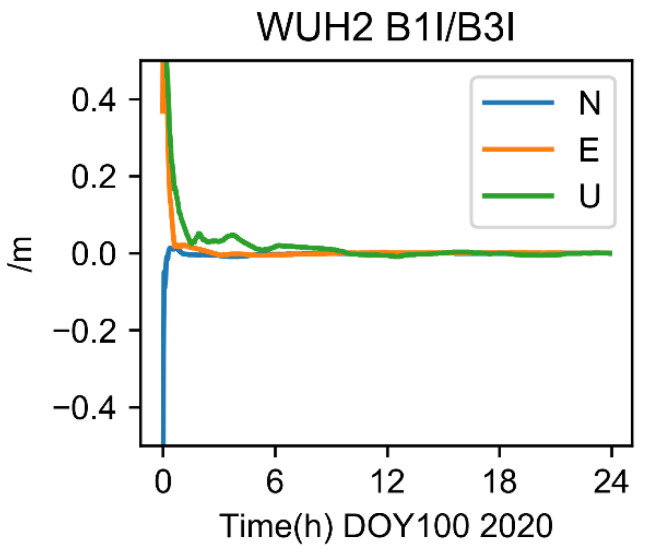
BDS B1I/B3I ionosphere-free results of WUH2.

**Figure 6 sensors-22-09059-f006:**
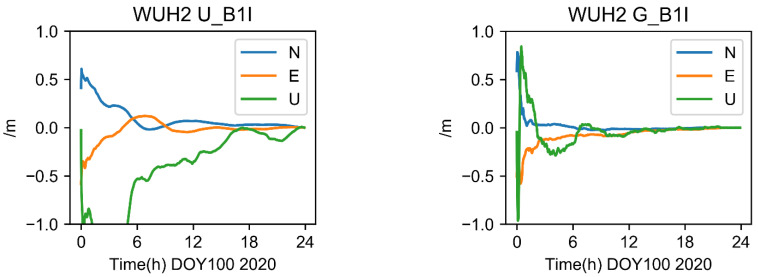
Positioning results of single-frequency algorithms at five frequencies (Strategy 1, Strategy 2).

**Figure 7 sensors-22-09059-f007:**
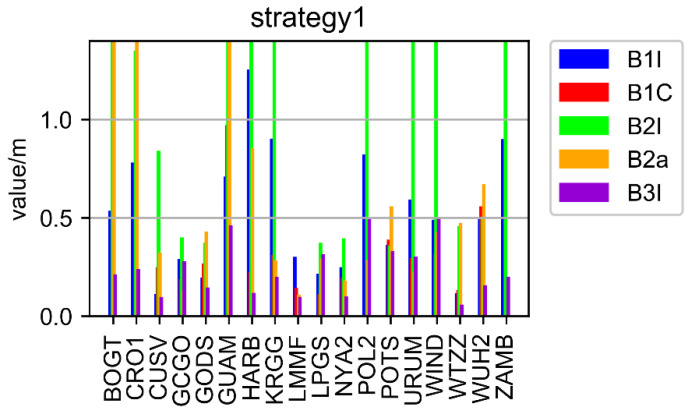
The results of Strategy 1.

**Figure 8 sensors-22-09059-f008:**
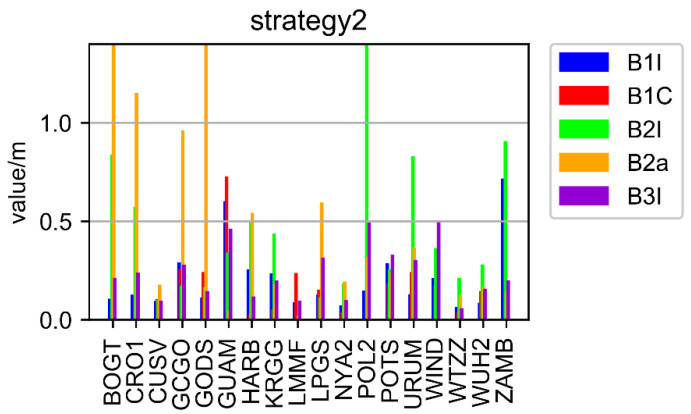
The results of Strategy 2.

**Figure 9 sensors-22-09059-f009:**
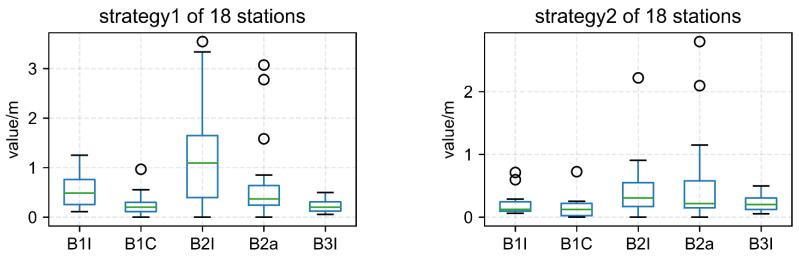
Average position value of 18 stations.

**Figure 10 sensors-22-09059-f010:**
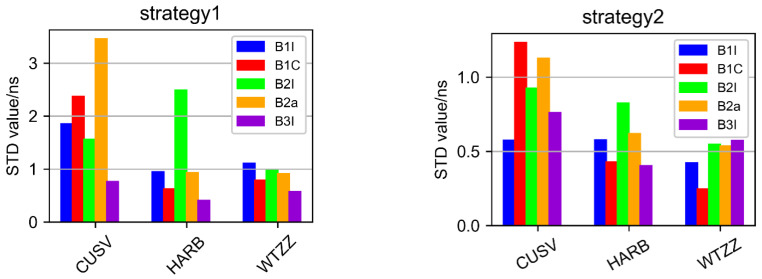
The results of clock difference statistics’ STD of two strategies of three stations (CUSV, HARB, WTZZ).

**Figure 11 sensors-22-09059-f011:**
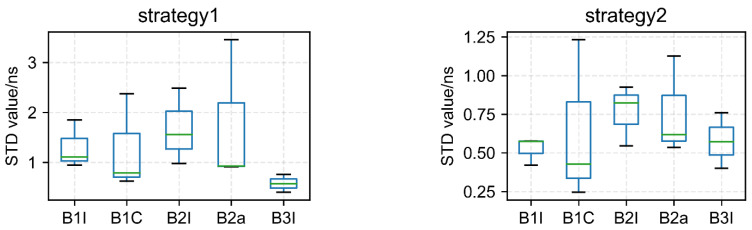
The average STD of clock difference of three stations with two strategies for different frequency points.

**Figure 12 sensors-22-09059-f012:**
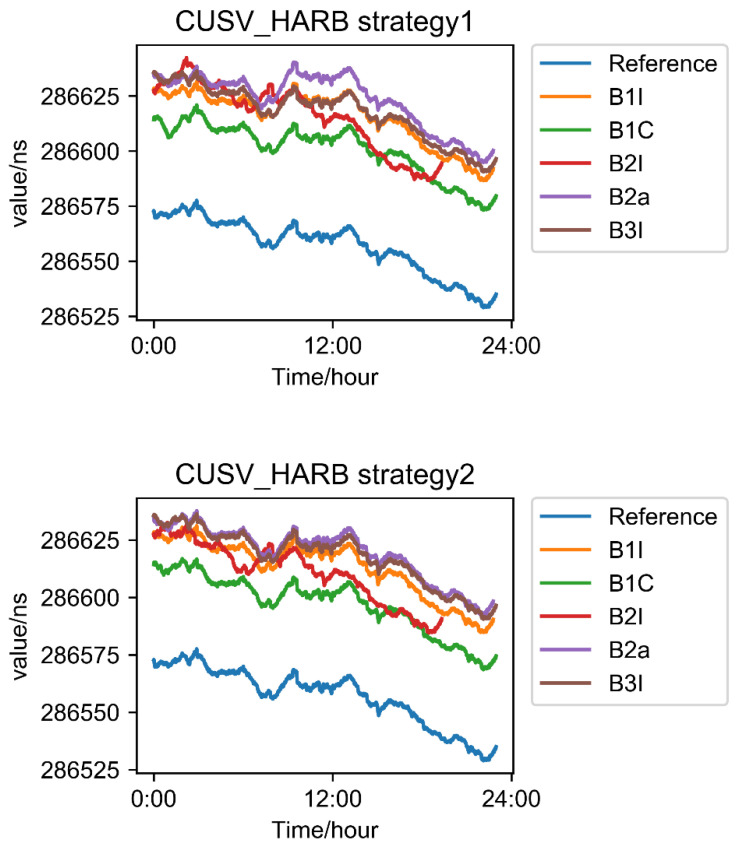
The time sequence diagram of two time links, two strategies, and five frequency points.

**Figure 13 sensors-22-09059-f013:**
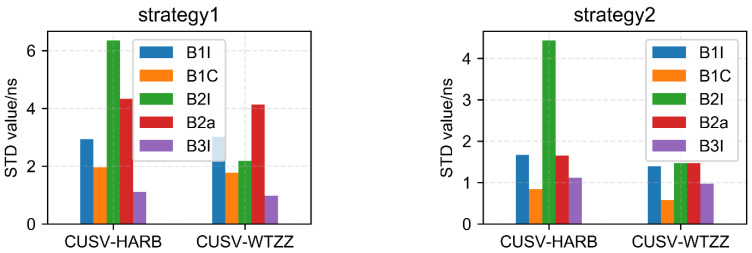
STD of two strategies of time links with five frequencies.

**Figure 14 sensors-22-09059-f014:**
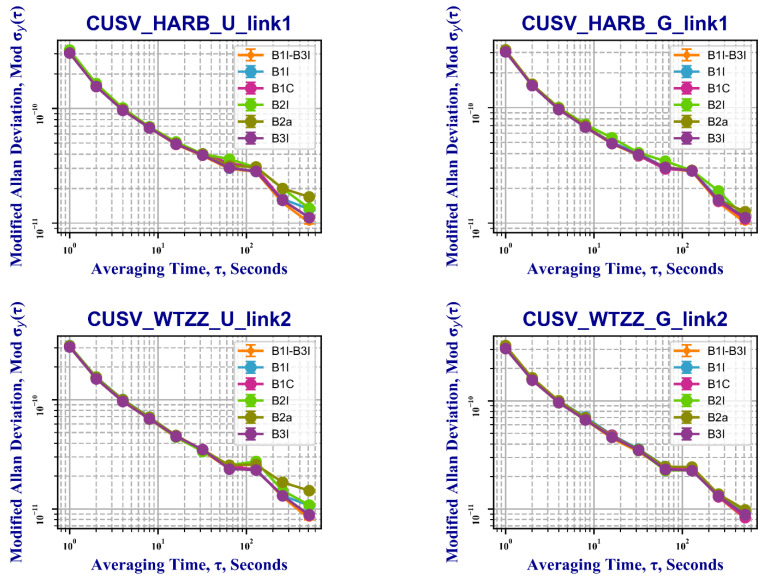
The frequency stability results of the two links of the two strategies (U, Strategy 1; G, Strategy 2).

**Table 1 sensors-22-09059-t001:** Signal characteristics of BDS-3 open service [13,14].

Frequency Band	Signal Component	Center Frequency	Modulation Mode	Information Rate/BPS	Compatible Interoperability
		(MHz)			
B1	B1C_data	1575.42	BOC(1,1)	50	GPS L1
	B1C_pilot		QMBOC(6,1,4/33)	0	Galileo E1
	B1I	1561.098	BPSK(2)	50(MEO/IGSO),500(GEO)	--
B2	B2a_data	1176.45	QPSK(10)	100	GPS L5
	B2a_pilot			0	Galileo E5a
	B2b_I	1207.14	QPSK(10)	500	Galileo E5b
	B2b_Q			500	
B3	B3I	1268.52	BPSK(10)	50(MEO/IGSO),500(GEO)	--

**Table 2 sensors-22-09059-t002:** Detailed strategy of PPP algorithm.

Parameter	Description
Wet tropospheric delay	UNB3 model [20]
Signal selection	(B1I, B1C, B2I, B2a, B3I)
Satellite antenna PCO and PCV	Igs14.atx
Elevation cut-off	7°
Solid earth tides	IERS Conventions 2010 [21]
Ocean loading	IERS Conventions 2010 [21]
Polar tides	IERS Conventions 2010 [21]
Phase wind-up effect	Wu model [22]
Receiver clock errors	Estimation, white noise
Station coordinate	Static model
Estimator	Kalman filter

## Data Availability

Not applicable.

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
