# Peer review of "A Modified Single-Frequency PPP Method for the Positioning and Time Transfer with BDS-3"

_sensors, 2022, doi:10.3390/s22239059_

Round 1
Reviewer 1 Report
The research done by Mingjun Ouyang is interesting and certainly of high practical interest, as most users of GNSS are using a single frequency channel.
I have very little to say about the contents, and the design of the study is pretty good.
My really complaint is about the command of the English language, that is quite poor in many places. I attach a scan of my annotated comments (with also minor scientific points to be clarified), as I have no time left to type them, but this point clearly needs to be fixed.
As an example, what is the meaning of "5 frequency"? It could be "five frequencies" or "frequency number 5", that carry a totally different meanings.

Author Response
"Please see the attachment."

Reviewer 2 Report
Review to Sensors-2014468 (A Modified single-frequency PPP method for the Positioning 2 and Time transfer with BDS-3).
To conduct the time transfer experiments for five types of single-frequency signals using the PPP method. Four open service signals (B1I, B1C, B2a, B3I) of the BDS-3 and the B2I signals broadcasted by the BDS-2 were used in this study. Compared with the conventional UC model, a modified GRAPHIC combination was proposed. The stability of BDS-3 single-frequency PPP is the same as BDS dual frequency PPP, the experiment results indicate that the BDS-3 single-frequency PPP can also be used as one of the formal methods for precision time comparison in the future. Thus, I would suggest a minor revision before it is considered for publication as following.
1. Adding references to the Modified GRAPHIC PPP model, I see that you cited but did not add references.
2. For the capitalized abbreviations that appear many times in the text, the complete words are not capitalized, such as: the root mean square (RMS) of the three-dimensional (3D). please check the full text.
3. Figure 13 Some occlusion, optimization recommended.
4. In the third conclusion at the end, it is suggested to highlight Beidou. As this article is all about Beidou, it is necessary to make clear the use of BDS-2 and BDS-3.
5. Please replot the figures in your paper (such as Figure 6, Figure 12 and Figure 14). It is best to include several subplots in one Figure.
Author Response
"Please see the attachment."
